# Full in-Office Guided Surgery with Open Selective Tooth-Supported Templates: A Prospective Clinical Study on 20 Patients

**DOI:** 10.3390/ijerph15112361

**Published:** 2018-10-25

**Authors:** Francesco Guido Mangano, Uli Hauschild, Oleg Admakin

**Affiliations:** 1Lecturer, Department of Medicine and Surgery, University of Insubria, 21100 Varese, Italy; 2Professor, Department of Prevention and Communal Dentistry, I.M. Sechenov First Moscow State Medical University, 119992 Moscow, Russia; admakin1966@mail.ru; 3Private Practice, 18038 San Remo, Italy; hauschild@dentaldesign.biz

**Keywords:** dental implants, guided implant surgery, template stability, template fit, complications, survival

## Abstract

Background: Guided implant surgery appears to have several benefits, such as the possibility of inserting flapless implants in a prosthetically driven manner, avoiding dangerous anatomical structures. However, to date, only a few surgeons routinely use guided surgery in partially edentulous patients. Aim: To present the results obtained with tooth-supported surgical templates characterized by an innovative open design with selective support, and manufactured via a full in-office procedure with a low-cost desktop 3D printer. Methods: Over a two-year period (2016–2018), all partially edentulous patients with one to three missing teeth (in maxilla and/or mandible), referred to a private dental practice for restoration with dental implants, were considered for inclusion in this prospective clinical study. An intraoral scanner (CS 3600^®^, Carestream Dental) and cone beam computed tomography (CS 9300^®^, Carestream Dental) were used to acquire the 3D information on the patients. Guided surgery software (SMOP^®^, Swissmeda) was used to plan the surgeries and to design open, selective, tooth-supported templates that were fabricated with a stereolithographic (SLA) desktop 3D printer (XFAB2000^®^, DWS). Guided implant surgeries were performed and patients were followed for a period of one year. The study outcomes were fit and stability of surgical templates, duration (time) of surgery, intra and post-operative complications, and implant stability and survival. Results: Twenty (20) partially edentulous patients (9 males, 11 females; mean age 54.4 ± 9.4 years) were included in the study; 28 open, selective, tooth-supported templates were designed with the aim of inserting 38 implants. Among the surgical templates, 24 had optimal fit and stability, three had optimal fit and sufficient stability, and only one had inadequate fit and unsatisfactory stability and was therefore not suitable for clinical use. The average time of the intervention was 15.7 ± 5.2 min per template. No intra-operative complications were reported, but one implant was not stable at placement and had to be removed. In total, 36 implants were restored with 10 two-unit fixed partial prostheses and 16 single crowns. All implants were successfully functioning at one year, even if, in two single crowns, minor prosthetic complications (abutment screw loosening) occurred. Conclusions: Full in-office guided surgery with open, selective, tooth-supported templates seem to represent a clinically predictable surgical procedure to restore partially edentulous patients. Further studies are needed to confirm these positive outcomes.

## 1. Introduction

To date, dental implants represent an effective and predictable solution for the prosthetic rehabilitation of partially and totally edentulous patients, with high survival and success rates in both the short and long term [1,2,3].

In the last ten years, the high predictability and profitability of dental implants have contributed to the diffusion of oral implantology around the world [4]. Consequently, implantology is no longer the prerogative of a few specialized professionals as it used to be, but has become a common technique manageable by almost all dental professionals [4].

Yet, the insertion of implants in the correct position, to the correct depth, and at the correct inclination remains rather complex, and implantology cannot be defined as risk-free for the practitioner [5].

In fact, serious consequences can follow upon the invasion of anatomical structures such as the inferior alveolar nerve or the maxillary sinus, both during the preparation of the implant site and during implant insertion—among them, paresthesias [6,7] and infections [8]. The invasion of the periodontal space of the adjacent tooth roots is itself an event to be avoided, which can determine the loss of vitality of the injured tooth and the failure of the implant [9]. Even more dangerous may be the perforation of the lingual bone wall in the anterior mandible [10], which can result in serious vascular lesions, which may endanger the patient’s life.

Beyond this, the incorrect positioning of an implant may involve aesthetic complications (especially in the anterior maxilla, where too buccal an insertion can result in marked bone resorption and aesthetic failure) [11,12] or serious difficulties in prosthetics, which result in biological and functional compromises [13]. Such compromises occur in cases of implants placed too close to the adjacent teeth, or too close each other, or with incorrect inclinations, which makes proper prosthetic restoration challenging for the dental technician [13].

Guided implant surgery today can represent the solution to all these problems [14,15,16]. Static guided implant surgery is a technique that consists of guided positioning of dental implants, by means of a surgical template, which transfers a computer-made three-dimensional (3D) planning process to the clinical context [14,15]. The implant is therefore inserted at the correct position, inclination, and depth as planned in the 3D software [14,15].

There are several advantages with guided surgery: in addition to controlling the implant position, depth, and inclination [15,16], there is the possibility of inserting implants flaplessly [17] (i.e., without having to raise a mucoperiosteal flap). This reduces post-operative pain for the patient and it simplifies the intervention. This minimally invasive approach allows the preservation of the periosteal microcirculation, with potential benefits to tissue stability [17,18].

Despite these benefits, however, to date, only a small fraction of surgeons routinely use guided surgery.

The reasons for this low use are to be found, above all, in the high production cost of the surgical templates, which are generally fabricated by external services of implant companies or specialized dental laboratories. For this reason, surgeons use these technologies only in the case of multiple implants and complex cases, in which the cost for the production of the guide is fully justified by the number of inserted fixtures. [16] A second and important factor that limits the use of guided surgery is the unsatisfactory accuracy of several surgical templates that are currently available on the market. Several literature reviews have in fact shown how a rather substantial deviation (on average, more than 1 mm) occurs between the planned position of the implants in the 3D software and their actual position after insertion [19,20].

In fact, many errors can occur, from image acquisition to implant planning, and from the production of the templates to the surgical procedure. These errors inevitably end up jeopardizing the final accuracy of the surgical templates [21].

Among these, one of the least considered but most important elements is that of the design and manufacture of surgical templates; in particular, an incorrect shape of the template does not allow one to obtain adequate stability, even in a case where powerful industrial 3D printers are used for the fabrication of the guides [22].

The aim of this prospective clinical study is to present the results obtained with tooth-supported surgical templates characterized by an innovative open design with selective support and manufactured via a full in-office procedure with a desktop 3D printer.

## 2. Materials and Methods

### 2.1. Inclusion and Exclusion Criteria

Over a two-year period (June 2016 to June 2018), all partially edentulous patients referred to a private dental practice (Studio Odontoiatrico Mangano^®^, Gravedona, Como, Italy) for restoration with dental implants were considered for inclusion in the present prospective clinical study. Inclusion criteria were: (1) Patients with one or two adjacent missing teeth in both arches (maxilla and/or mandible); (2) sufficient bone for the placing of implants at least 3.3 mm in diameter and 8.0 mm in length; and (3) patient’s willingness to participate fully in the protocol. Exclusion criteria were: (1) Patients with systemic diseases (e.g., uncontrolled diabetes, blood diseases, and psychiatric illnesses) that might represent a contra-indication to implant surgery; (2) patients undergoing chemotherapy and/or radiotherapy for cancer of the head and neck region; (3) patients receiving immunosuppressive therapies; (4) patients being treated with bisphosphonates (parenterally and/or orally); (5) patients with active periodontal infections (pus, fistulas, and periodontal abscesses); (6) patients with other oral diseases (vesiculobullous and ulcerative diseases, red and white lesions, and diseases of the salivary glands and cystic lesions); (7) patients with poor oral hygiene; and (8) patients with functional limitations or temporomandibular disorders. All patients were fully informed of the protocol of the present study and signed a detailed informed consent form prior to being enrolled. The study was conducted in full accordance with the principles outlined in the Helsinki Declaration on experimentation on human subjects (2008 revision) and was approved by the local Ethics Committee at Insubria University (protocol number #0034086-826).

### 2.2. Data Acquisition

Each of the patients selected for inclusion in the present prospective study was subjected to intraoral scan of the arches, with a powerful structured light scanner (CS 3600^®^, Carestream Dental, Atlanta, GA, USA) (Figure 1).

This scan was performed in orthodontic mode, and consisted of a scan of the master model (site of the edentulism), of the antagonist, and of the capture of the bite (occlusion). During the scan, the operator took care to capture every morphological detail of the teeth adjacent to the implant, i.e., those selected to support the future surgical guide. At the same time, special attention was paid to the scanning of soft tissues in the area of edentulism. The files thus acquired were saved in .STL format in a special folder. In the same session, the patient was then subjected to 3D radiological examination with cone beam computed tomography (CBCT), using a powerful scanner with adjustable field-of-view (CS 9300^®^, Carestream Dental, Atlanta, GA, USA) (Figure 2).

The operator was therefore able to select the more appropriate field of view (FOV) for the case (usually 10 × 5-cm FOV, to facilitate the overlap with the model derived from intraoral scanning, in the presence of metal artefacts, but in some cases without artifacts, even a 5 × 5-cm FOV) and then capture the patient’s 3D bone anatomy, which was immediately displayed in the CBCT reconstruction software. Particular attention was paid before the radiological examination to the centering of the area of interest, through specific dedicated tools; then, great attention was placed during the scan, so that the patient did not make any movement, and was completely immobile. After a first rapid visualization of the site of interest, and a first rapid measurement of the height and thickness of the residual ridge, confirming the feasibility of the case, the digital imaging and communication in medicines (DICOM) files derived from the CBCT were saved in a specific folder and were ready to be imported into the guided surgery software.

### 2.3. Planning

The planning of the case involved the use of two different sets of software: A prosthetic CAD (computer-assisted-design) (EXOCAD^®^, Darmstad, Germany) for the virtual wax-up and a surgical CAD (SMOP^®^, Swissmeda, Baar, Switzerland) for 3D planning of implant placement and the design of the surgical template (Figure 3).

In the order, the .STL files deriving from the intraoral scan were imported into the prosthetic CAD, where the virtual waxing of the restoration (single crown or partial fixed prosthesis) was carried out, taking into account the volumes, shapes, and interproximal contacts of the adjacent and occlusal teeth with the antagonist arch. The file of the waxing was thus saved in a special folder, always as .STL, in order to be opened in the surgical CAD before planning the implant placement. At this point, the operator opened the surgical CAD software by importing the DICOM files derived from the patient’s CBCT. The software presented the different reconstruction planes (axial, coronal, and sagittal) and allowed the importation of the intraoral scan file (.STL) that was superimposed on the reconstruction of bone from CBCT. The superimposition was performed with care, first for points and then for surfaces, and controlled in each section. This phase was the most delicate of all, since a possible overlap error can undermine the planning and therefore the positioning of the fixture in the exact desired position. Once the quality of the overlap between intraoral scan and bone reconstruction by CBCT was carefully checked and verified, the operator identified the anatomical structures of risk (alveolar inferior nerve, maxillary sinus) and the design of the panoramic curve, in order to obtain the desired cross sections. The operator imported the .STL file of the diagnostic wax-up and proceeded to plan an implant of a length and a diameter appropriate to the recipient site, taking into accounts the residual bone anatomy and the important information derived from the wax-up. The implant was therefore planned in the best possible position, and prosthetically guided in relation to the residual bone anatomy. Particular attention was paid to the position and inclination of the fixture and its compatibility with the prosthetic emergence. In the case of patients requiring treatment with more than one implant, the same operations were repeated for each of the selected fixtures. At this point, parametric instruments allowed the clinician to establish the distance between the sleeve and the implant shoulder, depending on the height of the available gingiva: In accordance, from the length of the selected implant, and from the height of the sleeve (5 mm in the present study) derived the data related to the drilling depth, a key element in the surgical phase. Furthermore, the parametric tools of the software made it possible to adjust the width of the outside diameter of the hole in the template, for the insertion and fixation of the sleeve.

Since, in this study, the sleeves used were characteristic of an outer diameter of 6 mm, a certain degree of tolerance was established (0.07 mm) so that the diameter of this hole was fixed at 6.07 mm. After setting these key elements, the position of the implants was blocked and the operator switched to the surgical template design. A few simple CAD operators allowed the surgeon to draw an open selective tooth-supported surgical guide. The support was based on buccal and palatal (lingual) clamps and selective occlusal supports. The surgeon was free at this stage to regulate the degree of undercut of the clamps, and their size, as well as the dimensions and features of the occlusal and mucosal supports. This design phase was carried out in a simple but parametric way, allowing the surgeon to have total dimensional control of the components and shapes drawn. The peculiarity of these surgical templates was demonstrated by the fact that they were open, with selective supports on the dental surfaces, controllable in section, for a perfect verification of the adaptation on 3D-printed models and constant intraoral control during surgery. At the end of this process, the software calculated a surgical guide that was easily exported as an .STL file, upon payment of an export fee (with a variable price, in relation to the number of planned cases and exports requested).

### 2.4. 3D Printing of the Models and the Surgical Templates

After having modeled and saved the surgical templates, the clinician provided for the printing of 3D models derived from intraoral scanning (Figure 4), and of the same surgical templates (Figure 5), with a powerful stereolithography (SLA) desktop printer (XFAB2000^®^, DWS, Thiene, Vicenza).

In particular, two different resins were used: the Invicta917^®^ anthracite (gray) for the production of the physical models of the patient’s dental arches, and the DS3000^®^ (transparent) for the production of surgical templates. For printing of the models and surgical templates, the .STL files were imported into a 3D printing planning software (Nauta^®^, DWS, Thiene, Vicenza) and positioned on the printing plate. Within this software the pins and support for 3D printing were inserted, with defined parametric features, and a base was introduced for a better stabilization of the model during printing. Care was taken to avoid pins and supports being positioned at the level of the occlusal table of the models (and in particular at the level of the support points of the surgical guide, which had to remain free) or, even worse, at the internal surface (support) of the surgical templates. Once the 3D printing was planned, the models and the surgical guides could be phisically reproduced, thanks to 3D-printer management software (Fictor^®^, DWS, Thiene, Vicenza). Within this software, it was possible to decide the material with which to print the models and the surgical templates (the materials were provided in different interchangeable trays) and to control the functionality of the printer in a simple and intuitive way. At the end of each printing process, the final product (dentate model or surgical template) was then thoroughly rinsed in two ethyl alcohol baths, for 10 min each; as a result of this, the sleeves were positioned and blocked inside the surgical templates for clinical use. The surgical templates were then manually adjusted on 3D-printed models, taking care to eliminate excessive undercuts and to obtain an absolutely stable fit. This adaptation took place by fitting the surgical templates from 8 to 10 times on the corresponding toothed models. At the end of this procedure, the surgical templates were sterilized and were ready for use in surgery.

### 2.5. Surgery

The patient sat on the dental chair (Figure 6) and rinsed 3 or 4 time with chlorexidine 0.2% mouthwash, 10 min before surgery, for a total of 2 min.

After local anesthesia by infiltration with articaine with adrenaline (at a ratio of 1:100,000) the surgeon positioned the dental-supported surgical guide, verifying its stability and the exact adaptation on the occlusal surfaces of the teeth (Figure 7).

This verification was possible thanks to the design of the surgical guide, which was open and thus allowed the surgeon to check directly the perfect fit of the template on the occlusal surfaces of the teeth. Once the adaptation of the template was verified, according to the surgical plan, the surgeon made the mucotomy, with a dedicated mucotome, through the surgical guide, in order to remove the necessary amount of gingiva and to create an access for the preparation of the implant site. Once the mucotomy ended, the surgeon began to prepare the implant site (Figure 8), in accordance with the surgical plan (and therefore with the pre-established drilling depth) and consistent with the chosen implant diameter and length.

The implants used in this study were conical with aggressive thread design (BT Safe Bone Level^®^, BTK, Dueville, Vicenza, Italy), available in different diameters (3.3, 3.75, 4.1, and 4.8 mm, respectively) and lengths (6, 8, 10, 12, and 14 mm, respectively). The preparation of the surgical site was as follows. Once the mucosal operculum was opened, the surgeon took the pilot drill (2.0 mm) and mounted on it the selected stop, according to the surgical plane (depth stop, to be able to drill exactly to the desired depth). Once this operation was completed, the surgeon mounted the reducer of the diameter corresponding to the drill (2.0 mm) inside the sleeve. At this point, the surgeon drilled up to the stop, that is, bringing the stop of the drill in contact with the sleeve shoulder, under abundant irrigation of saline solution. The same operations were repeated for the different incremental diameter drills. In fact, the surgeon removed the 2.0-mm reducer and replaced it, in the sleeve, with a 2.5-mm reducer. The 2.5-mm drill was mounted on the handpiece with the depth stop previously used and the surgeon drilled up to the stop. These steps were repeated until the implant site was prepared at the desired diameter, through the passage of incremental diameter drills (3.1, 3.45, 3.75, and 4.35 mm). In particular, in order to be able to position an implant 3.3-mm in diameter, the last drill was 3.1 mm in diameter; in order to be able to place a 3.7-mm implant, the last drill was 3.45 mm; to be able to position a 4.1-mm implant, the last drill passed was 3.75 mm; finally, in order to place a 4.8-mm implant, the last drill was 4.35 mm in diameter. Once the implant bed was prepared, the surgeon selected the fixture of the desired diameter and length and inserted it through the surgical guide and then through the sleeve (without reducer) (Figure 9).

The implant was initially inserted through the handpiece, set with a maximum insertion torque of 30 Ncm; exceeding this threshold, the surgeon proceeded manually for better control. At the end of the insertion, the surgeon proceeded to remove the template and check the implant axis with a dedicated instrument, then placed a transmucosal healing screw of such height as not to be covered by mucosa during the healing phase (Figure 10).

The operation was completed and the patient was left to go home with the indication to perform 3 or 4 mouthwashings with chlorexidine 0.2% within 4 days after the operation, and with a prescription of painkiller to be taken when needed (ibuprofen 600 mg, maximum 2/day). Since no sutures were placed, the first follow-up visit was set at 10 days after the intervention.

### 2.6. Prosthesis

The prosthetic phases started with the removal of the transmucosal healing abutment, 2 months after implant placement, and an optical impression with an intraoral scanner (CS 3600^®^, Carestream Dental, Atlanta, GA, USA) after scanbody positioning (Figure 11).

Scanning was limited to the sector of interest, including the antagonist arch. The impression (.STL or .PLY file) was then sent to the dental technician, who proceeded with the design of the provisional polymethylmethacrylate (PMMA) prosthetic restoration and the individual customized zirconia abutment in computer-assisted-design (CAD) software (DentalCAD^®^, Exocad, Darmstadt, Germany) (Figure 12).

The individual customized abutment was designed to be cemented on a titanium abutment (BT link^®^ BTK, Dueville, Vicenza, Italy), screwed on the implant. The individual customized abutment was milled in zirconia with a powerful 5-axis milling machine (DWX-50^®^, Roland, Ascoli Piceno, Italy), sintered in an oven (Tabeo^®^, Mihm-Vogt, Stutensee, Germany), and cemented extra-orally on the titanium base. The provisional crown, however, was produced by milling PMMA with a 4-axis milling machine (DWX-4^®^, Roland, Ascoli Piceno, Italy). After the extra-oral cementation, the individual hybrid (titanium/zirconia) abutment was positioned and screwed onto the implant. The PMMA provisional restoration was finally cemented on it with a zinc-oxide eugenol cement (TempBond^®^, Kerr, Orange, CA, USA) (Figure 13).

Occlusion was carefully checked and then polishing and staining/characterization was done. Any occlusal precontacts recorded in this phase with articulating papers (Bausch Articulating Paper^®^, Bausch Inc, Nashua, NH, USA) were photographed, in order to subsequently guide the modeling of the final restoration. The PMMA provisionals were left in place for a period of 2 months; after that, they were replaced with definitive monolithic zirconia restorations. In all cases, translucent zirconia (Katana^®^, Kuraray Noritake, Tokyo, Japan) was employed. The design of the final restorations was obtained by properly modifying the design of the provisions in the previous CAD scene, without the need of other impressions. Care was taken to adapt the cement spaces to the needs of the new material (zirconia) and to check and modify the occlusal contact points, based on the indications collected at the time of positioning of the PMMA provisionals. The final restorations were cemented with zinc-oxide eugenol cement (Figure 14).

In all cases, no physical models of the jaws were prepared.

### 2.7. Study Outcomes

Patients were followed for a minimum period of 1 year from implant placement, with a minimum of 3 or 4 annual follow-up visits from implant placement. The outcomes of the study were different and collected both directly at the time of surgery (stability and adaptation of surgical templates, time of surgery, intraoperative complications related to the guided surgery procedure, stability of the implant), and after a time, during the annual check-ups (biological and prosthetic complications) and at the end of the study, that is, at the 1-year follow-up control (implant survival).

#### 2.7.1. Fit and Stability of Surgical Templates

The fit and stability of the surgical templates represented the first outcomes of the present study. Fit and stability of the guides were clinically verified by the surgeon at the time of surgery. The fit, or adaptation of the template, essentially depended on the fit of the guide on the occlusal surfaces of the supporting teeth and on the approximal surfaces of the adjacent teeth. For this reason, the fit was first verified through a careful visual analysis of the adaptation of the template on the occlusal surfaces of the supporting teeth. This analysis was possible thanks to the particular design of the surgical templates used in this study, which allowed one to evaluate the adaptation of the occlusal supports directly in the section, and therefore the exact correspondence between the design of the surgical guide and the anatomy of the occlusal table. The fit then also depended on the adaptation of the template on the approximal surfaces of the teeth adjacent to the implants: This adaptation had to be perfect and did not have to require further interventions or modifications during the operation. According to the protocol of the study, having tested the surgical guide in the patient’s mouth, the surgeon could define the fit of the template as optimal, sufficient, or inadequate. In a case of optimal fit, no retouching was necessary and the surgical guide was used without any modification, as received after sterilization. In case of sufficient adaptation, the template required some minor modification (such as polishing of one or more surfaces) to be adapted to the best and proceed with the intervention. These adaptations had to be minimal, because they could potentially affect the position, inclination, and depth of the implant and thus generate a discrepancy between planning and actual position of the implant after surgery. In this case, however, the fit was inadequate (and required major changes to be able to use the template), the surgeon could decide to eliminate the surgical guide and proceed with classic surgery, raising a full-thickness mucoperiosteal flap and manually preparing (without any guide) the implant site. Likewise, the stability of the surgical guide (which depended not only on the fit on the occlusal surfaces of the abutment teeth and on the approximal surfaces of the teeth adjacent to the implant, but also on the retentive clamps positioned at the collar of the supporting teeth) was clinically verified by the surgeon at the time of the intervention. A template was considered stable if equipped with a perfect fit, and absolutely immobile during all phases of surgery (insertion of reducers in the sleeves, drilling, etc.). In the case of an optimal stability, the template offered a certain resistance to the insertion, making a snap that allowed it to be blocked perfectly by the clamps, thanks to a minimum amount of undercut designed to perfectly stabilize the assembly. If the template had a slight movement of swinging/jiggling, the surgeon could proceed, but the stability of the template was defined as sufficient. In a case of insufficient stability, with a major oscillation movement, the surgeon could suspend the procedure and, as explained above, eliminate the surgical guide and proceed with classical surgery, with a full-thickness flap and manual preparation (not guided) of the implant site. In all the cases performed, fit and stability of the surgical templates were reported in the patient’s record.

#### 2.7.2. Duration (Time) of the Surgery

From the anesthesia and the trial of the template to the positioning of the healing screw and the removal of the template, the chair assistant monitored the exact time taken by the surgeon. This time, measured in minutes, was marked on the patient’s record.

#### 2.7.3. Intraoperative Complications Related to the Guided Surgery Procedure

Any complication encountered during the operation (not only insufficient fit or instability of the template) was marked in the patient’s record folder and reported in the present study. Among the intraoperative complications related to the procedure of guided surgery were: Fracture of the surgical guide; insufficient opening of the mouth by the patient, making it impossible to prepare the implant site through the surgical guide; and a clearly wrong insertion of the implant (in terms of position, inclination, and depth) that required opening of a mucoperiosteal flap, removal of the implant, and new preparation. Invasion of anatomical risk structures (inferior alveolar nerve and maxillary sinus; adjacent teeth) and perforation of the buccal and palatal (lingual) bone corticals represented the most insidious complications related to errors (planning or execution) during the guided surgery procedure, and they were included in the patient’s record.

#### 2.7.4. Clinical Stability of the Implant

The stability of each fixture was checked clinically by applying a reverse torque of 20 Ncm. The stability was checked three times: immediately after placement, at the delivery of provisional, and at final restorations.

#### 2.7.5. Post-Operative Complications

The complications that could occur in the two weeks following the operation were classified as post-operative complications. They included pain, discomfort, exudation and suppuration, swelling, and infection of the implant.

#### 2.7.6. Complications during Follow-Up

All the complications that could affect the implants from the second week of surgery until the end of the study, as well as all the prosthetic complications that affected the implant-supported restorations, were marked in the patient’s record. These complications were divided into biological and prosthetic ones. Biological complications included peri-implant mucositis and peri-implantitis. Among the prosthetic complications were screw loosening and/or fracture, as well as fracture of the zirconia abutment/restoration.

#### 2.7.7. Implant Survival

An implant was considered ‘‘surviving’’ if it was still functioning at the last clinical control. An implant was, conversely, considered to have ‘‘failed’’ in all cases in which the clinician was forced to remove the fixture because of: Lack of osseointegration with fixture mobility, in the absence of infection; severe and/or recurrent infection (peri-implantitis) with marked bone loss; progressive bone loss in the absence of infection, but with implant mobilization; and fracture of the implant. The failures could be defined as ‘‘early’’ if they occurred before the prosthetic restoration; they were defined as ‘‘late’’ if they occurred after placement of the prosthetic abutment and functionalization of the implant with the provisional crown.

#### 2.7.8. Statistical Analysis

Data was collected by an independent operator who was not directly involved in the placement of fixtures. Descriptive statistics were performed for the patients demographics (gender, age at surgery, and smoking habit) and the features of the inserted implants (site, position, length, diameter). Absolute and relative frequency distributions were calculated for qualitative variables (fit and stability of the surgical templates, implant stability and survival, complications) while means, standard deviations, medians, and 95% confidence intervals (CI) were estimated for quantitative variables (patient’s age at surgery, duration/time of the surgery). Implant stability, survival, and the incidence of complications were calculated at the patient level and at the restoration level.

## 3. Results

In total, 20 partially edentulous patients (9 males and 11 females, mean age 54.4 ± 9.4 years, range 39–67 years, median 55.5 years, confidence interval 95% 50.3–58.5 years) were considered eligible for enrollment in the present prospective clinical study. In total, 28 open, selective, tooth-supported templates were designed, with the aim of inserting 38 internal hexagonal implants (BT Safe bone level^®^, BTK, Povolaro di Dueville, Italy). Among the selected patients, only 4 (4/20: 20%) were smokers and 2 (2/20: 10%) were bruxists. Among the planned implants, 19 (19/38: 50%) were maxillary and 19 (19/38: 50%) were mandibular; in particular, three of the planned implants (3/38: 7.9%) were incisors, one (1/38: 2.6%) was a cuspid, 20 (20/38: 52.6%) were premolars, and 14 (14/38: 36.9%) were molars. With regard to the length of the planned fixtures, 14 (14/38: 36.9%) were 8.0 mm long, 18 (18/38: 47.3%) were 10.0 mm, and six (6/38: 15.8%) were 12.0 mm. Finally, with regard to implant diameter, seven (7/38: 18.4%) were 3.3 mm in diameter, 14 (14/38: 36.9%) were 3.75 mm, nine (9/38: 23.7%) were 4.1 mm, and eight (8/38: 21.0%) were 4.8 mm.

Among the surgical templates, 24 exhibited optimal fit and stability, three had an optimal fit but only sufficient stability, and one had an inadequate fit and unsatisfactory stability. In the case of the templates with optimal fit and stability, the surgeon proceeded with the intervention, as well as in all 3 cases in which the stability was slightly lower (only sufficient) and the fit optimal. In the latter cases, however, the surgeon was helped by the assistant who kept the surgical templates in the correct position by exerting a certain pressure with two fingers. In the case in which the stability and the fit were not satisfactory, the surgeon elevated a full-thickness flap and proceeded with conventional (and not guided) implant placement. The reason for this failure due to imperfect adaptation was identified in the long time passed between the manufacture of the template and the surgery (over a month of waiting) with the possibility of a deformation of the resin template and slight movements of the supporting teeth.

Regarding the timing of surgery, in the 19 patients (27 templates) actually treated with guided surgery, the average time of the intervention (from the anesthesia to the placement of the healing screw) was 15.7 ± 5.2 min per template (range 8–25 min, median 15, confidence interval 95% 13.8–17.6) and 11.4 ± 2.9 min per implant (range 7–18 min, median 11 min, confidence interval 95% 10.5–12.3). The average time required was slightly higher in the case of non-optimal and only sufficient stability of the templates (13.6 ± 1.5 min per implant).

No intraoperative complications related to the guided surgery and to the flapless approach were reported in the present study: therefore, no guides fractured, no insufficient patient opening, and no major errors in the position, inclination, and depth of the implants were evident. No anatomical risk structures (inferior alveolar nerve, maxillary sinus) were invaded and no cortical bone perforations occurred.

At placement, the stability of all 37 implants placed through the guides with a flapless approach was acceptable, except in one case (a first mandibular molar in a 47-year-old female smoking patient) in which the receiving site was not completely healed and the fixture was not stable. This fixture was removed and the surgeon had to wait for a few months, before proceeding with the placement of the implant via a conventional, non-guided approach. No issues were reported for the stability of all other 36 implants during the entire follow-up of the study, or at the subsequent controls (at the delivery of provisional and final restorations).

No post-operative complications were reported after surgery; all patients in fact benefited from the minimal invasiveness of the flapless approach and no one complained of pain or swelling in the first two weeks after surgery. No exudation nor suppuration, swelling, or infections were reported. The incidence of immediate post-operative complications was 0% (0/36 implants).

The 36 implants were successfully restored with 10 two-unit fixed partial prostheses and with 16 single crowns.

During the follow-up period and the provisionalization, two single crowns underwent a minor prosthetic complication, such as abutment-screw loosening. However, the abutments were screwed on the implants again and no further mechanical complications were reported for these fixtures. At the end of the study, the incidence of complications was 7.6% (2/26 restorations), but these complications were limited at the period of the provisionals.

At 1 year year after placement, all 36 implants were regularly functioning, for a survival rate of 100% (36/36 implants surviving) (Figure 15).

The implant distribution with related survival and complication rates are summarized in Table 1.

## 4. Discussion

In recent years, the digital revolution has radically changed the world of dentistry [23]. In fact, powerful devices have been introduced, such as intraoral scanners, able to acquire an extremely accurate dental arch impression with the use of light [24], and cone beam computed tomography (CBCT), thanks to which the clinician can have precise three-dimensional (3D) information on height and thickness of the patient’s residual bone crest, with a low radiation dose [25]. These machines, together with the surgical and prosthetic CAD software, allow the clinician to work in a predictable manner, within a completely digital workflow [24,25,26].

The ideal application area for these technologies seems to be implantology. In fact, thanks to CBCT and intraoral scanners, it is now possible to acquire accurate information on bone, dental, and soft tissue anatomy; combining this information in a virtual environment, within dedicated surgical CAD, it is possible to construct a virtual patient model [27] on which to plan the intervention (for example, the placement of implants [14,15,28], but also a bone regeneration with custom-made grafts [29,30], blades [31], meshes [32], as well as root analogue [33] or subperiosteal [34] implants).

In the case of guided implant surgery, the procedure is rather simple, because immediately after planning, thanks to a few parametric and modeling tools, the available CAD software allows one to model a surgical template [14,15,28,35]. This template will be produced by 3D printing [22,36] or by milling [37] and used clinically for guided implant placement.

Although the advantages of the guided approach in implantology are obvious—the possibility of inserting fixtures at an ideal position, inclination, and depth, “prosthetically guided”, and therefore avoiding dangerous anatomical structures (inferior alveolar nerve, maxillary sinus, adjacent teeth) [11,15,28,35,38]—to date, few professional clinicians have routinely adopted guided surgery in their clinical practice. The reason for all this must certainly be sought in the rather high costs of the systems originally available, in their complexity, and in the need to use external services for the manufacture of the templates.

Today however, things have changed, and with an intraoral scanner, a CBCT, simple CAD software with few parametric modeling operators [39], and a desktop 3D printer, the clinician can easily plan his/her surgeries within the dental clinic, without having to use expensive external services.

In the present prospective clinical study, 20 partially edentulous patients (9 males, 11 females; mean age 54.4 ± 9.4 years) were included. In total, 28 open selective tooth-supported templates were designed, with the aim of inserting 38 implants. Among the surgical templates, 24 exhibited optimal fit and stability, 3 had optimal fit and sufficient stability, and only one had inadequate fit and unsatisfactory stability and was therefore not suitable for clinical use. Since one template (for one single implant) was not suitable for clinical use, 37 implants were inserted in 19 patients through open, selective, tooth-supported templates. The average time of the intervention was 15.7 ± 5.2 min per template. No intra-operative complications were reported, but one implant was not stable at placement and had to be removed. In total, 36 implants were restored with 10 two-unit fixed partial prostheses and with 16 single crowns. All implants were successfully functioning at one year, even if in two single crowns, minor prosthetic complications (abutment screw loosening) occurred.

The most interesting result obtained from our present study is the high stability and fit of the surgical templates. While printed in 3D directly in the office, using a low-cost desktop 3D printer, in fact, our surgical templates were extremely stable and had an optimal fit. This is essential for extending the applications of guided surgery even to the partially edentulous patients, up to the single tooth gap, with tooth-supported templates. Until now, the most frequent application of the guided surgery has been represented by totally edentulous patients [40,41]. The reason for this excellent result has to be found in the particular type of design of the guides that the SMOP^®^ CAD software allows one to realize, with open, selective, tooth-supported surgical templates (supported by points and not by surfaces) [35,39,42]. Supporting the guide selectively by points allows one, in fact, to reduce the negative consequences that scanning errors may have. Moreover, the fact that the templates are open allows the surgeon to control the optimal fit of the guide on the occlusal surfaces directly in the mouth [35,39,42]. This design is optimal and represents a huge advantage compared with other software, which draw conventional closed templates supported by surfaces; in such cases, a single scanning error or a printing imperfection, in fact, may result in instability of the template, which will point on some surfaces and will basculate [14,15,21]. A completely closed template also does not allow the control of the fit on the occlusal contacts, and forces the clinician to work “blindly”. Last, but not least, completely closed and surface-supported guides require very powerful 3D printers for fabrication, to try to reduce the effects that printing errors can have on the final stability of the template. In the present study, on the other hand, we showed how, with an open template design with selective supports, it is possible to produce high-stability surgical templates directly in the office and at low cost, without having to resort to external services.

The printer used in this study was an XFAB2000^®^, produced by DWS, characterized by a proprietary SLA technology, with extremely accurate and precise BluEdge^®^ laser, calibrated to obtain smooth surfaces that do not require post-production. The printer has a cylindrical working area of 180 mm in diameter, and can use 12 proprietary resins; in relation to them, the thickness of the layer is variable and can reach up to 60 microns. The printer is equipped with parametric software (Nauta^®^) for the automatic generation of the supports necessary for printing. Thanks to this software, the clinician can quickly draw the supports and optimize them according to his/her needs. The software allows ample freedom to modify the supports and the orientation and position of the objects to be printed in 3D, so as to favor the best quality of the surfaces of the output. The file obtained is then sent to the Fictor^®^ proprietary software, which starts and manages the print operations.

The greater stability of the surgical templates naturally has the potential to improve the accuracy of guided surgery, i.e., the correspondence between the planned implant position and the actual implant position after surgery [43]. To date, several studies and literature reviews have shown that, with conventional, closed, and surface-supported surgical templates, there can be a consistent deviation between the planned and the actual position of implants after surgery, even greater than 1 mm both at collar and apex of the fixture [43]; the same angular deviations are consistent [44]. The development of increasingly powerful software and the advent of intraoral scanning allow one today to more accurately investigate the linear and angular deviations between the planned and actual position of the implants [45]. In fact, it is no longer necessary to overlap two CBCTs (rather inaccurate and antiquated technique), but it is sufficient to extract the planned implant position from the guided software, as .STL files, and superimpose it on the actual (real) position of the implants, captured by an intraoral scan with a scan body in position. The SMOP^®^ software implements this function, thanks to dedicated libraries that also contain the implant scan bodies. Studies are underway to check whether the 3D deviations between the planned position and the actual (real) position of the fixtures after surgery can really be reduced, relative to what has been reported in the literature until now.

The present study has limitations, in fact, although prospective in design, it bases its conclusions on a relatively small number of enrolled patients, manufactured templates, and inserted implants. Further prospective studies on a larger number of patients will be needed to confirm the good results obtained in this work. Furthermore, in the present study, patients with functional limitations and reduced opening of the mouth were preliminarily excluded; it is known that in such patients, the size of the surgical templates and the necessity of using long drills can represent an issue, especially in the posterior areas (first and second molars, upper and lower jaws). The parametric functions of the SMOP^®^ software allow one, in part, to limit the need for vertical space, by moving the sleeves toward the soft tissues as much as possible [35,39,42]; however, the problem of vertical space remains and will always be actual in guided surgery, until sleeves can be used to guide the drills. The sleeve, in fact, “steals” at least 5 mm of space to be able to drive the drills effectively, and this limitation forces the clinician to use long drills, incompatible with the treatment of the posterior areas of some patients. In addition, the sleeve reduces visibility and interferes with proper bone irrigation, necessary to prevent overheating and the negative consequences that may result from it, during the preparation of the surgical site.

## 5. Conclusions

In the present prospective clinical study on 20 patients, the use of open, selective, tooth-supported templates fabricated with a desktop SLA 3D printer was clinically successful, as all the surgical templates (except one) exhibited excellent fit and stability, making the flapless placement of 37 implants possible without any intra-operative complication (with the exception of one fixture that did not have sufficient clinical stability at placement and had to be removed). One year after placement, all implants were successfully functioning and no post-operative complications were registered, with the exception of two single crowns that had their abutments loosened (these abutments were screwed-in again and no further prosthetic complications occurred). Hence, full in-office guided surgery with open, selective, tooth-supported templates seems to represent a clinically predictable surgical procedure to restore partially edentulous patients. Further studies are needed to confirm these positive outcomes.

## Figures and Tables

**Figure 1 ijerph-15-02361-f001:**
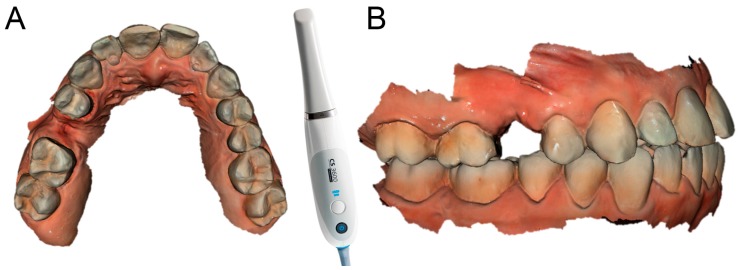
Intraoral scan CS 3600^®^ (Carestream Dental, Atlanta, GA, USA). The second right premolar (#15) is missing and an implant is going to be planned. (**A**) Occlusal view. (**B**) Lateral view.

**Figure 2 ijerph-15-02361-f002:**
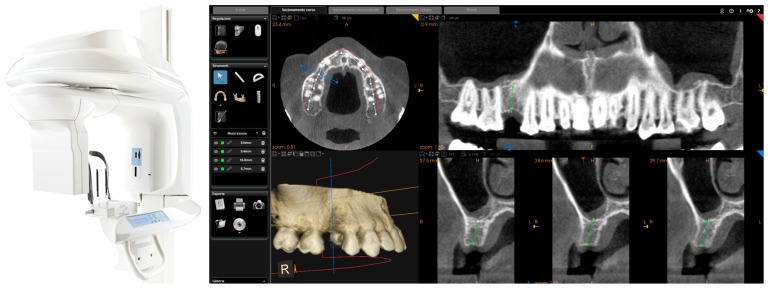
CBCT with CS 9300^®^ (Carestream Dental, Atlanta, GA, USA). The first visualization of the CBCT reveals sufficient bone volume (height and width) to plan the position of a dental implant.

**Figure 3 ijerph-15-02361-f003:**
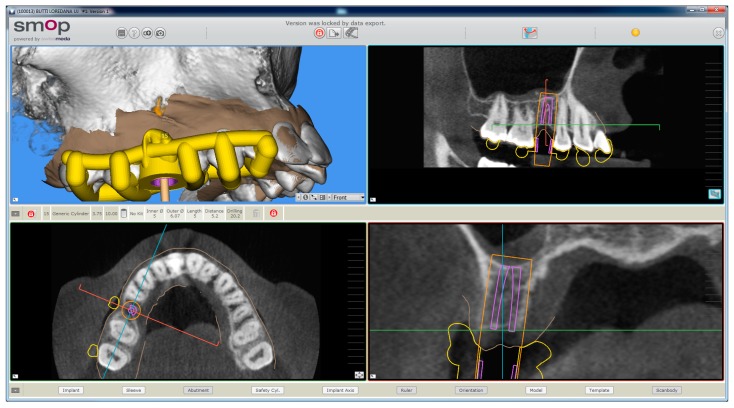
The implant is planned with the aid of a guided surgery software (SMOP^®^, Swissmeda, Baar, Switzerland). A 3.75 mm × 10 mm implant (BT Safe Bone Level^®^, BTK, Dueville, Italy) is therefore planned.

**Figure 4 ijerph-15-02361-f004:**
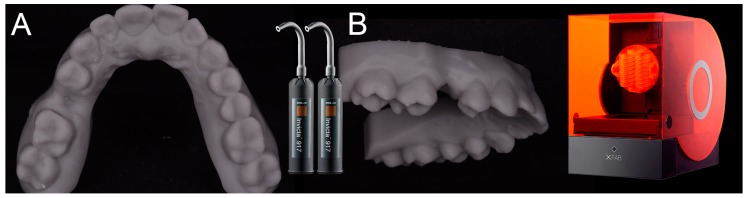
The models are printed in resin (Invicta917^®^) with a desktop stereolithographic (SLA) 3D printer (XFAB2000^®^, DWS, Thiene, Italy). (**A**) Occlusal view. (**B**) Lateral view.

**Figure 5 ijerph-15-02361-f005:**
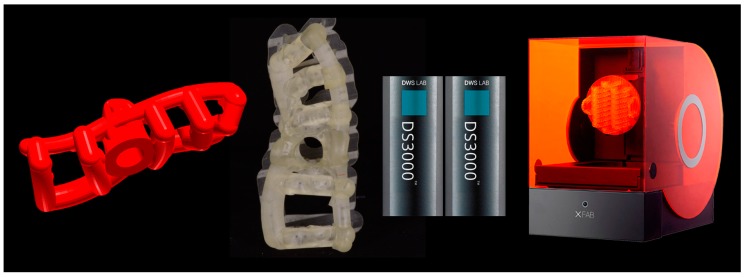
The open selective tooth-supported surgical template is printed with the XFAB2000 SLA printer (XFAB2000^®^, DWS, Thiene, Italy) using the DS3000^®^ resin.

**Figure 6 ijerph-15-02361-f006:**
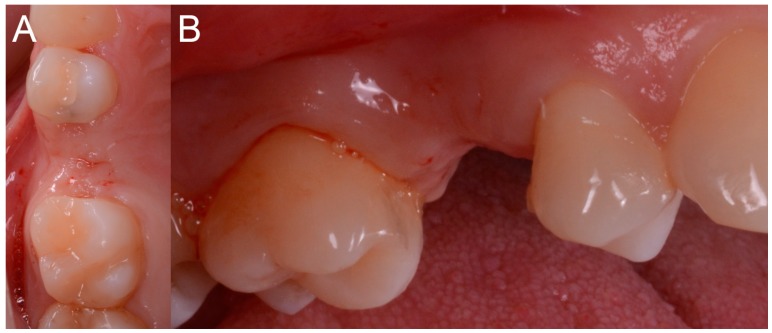
Pre-operative situation. (**A**) Occlusal view. (**B**) Lateral view.

**Figure 7 ijerph-15-02361-f007:**
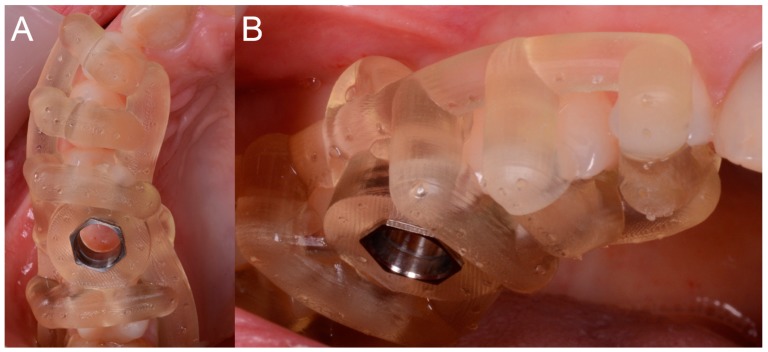
The surgical guide in position with the sleeve inserted in. (**A**) Occlusal view. (**B**) Lateral view.

**Figure 8 ijerph-15-02361-f008:**
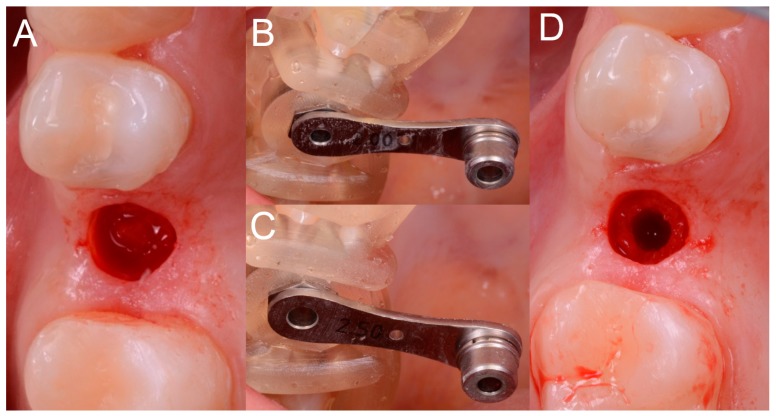
Different phases of surgery. (**A**) Mucotomy; (**B**,**C**) Preparation of the implant bed with different reducers inserted in the sleeve. (**D**) Occlusal view during preparation of the implant bed.

**Figure 9 ijerph-15-02361-f009:**
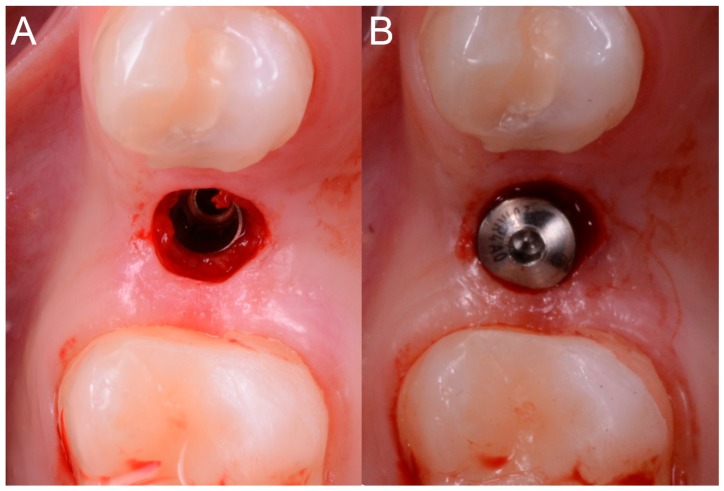
Implant positioning. (**A**) The implant (BT Safe Bone Level, BTK, Dueville, Italy) is inserted through the guide, in the planned position, inclination and depth; (**B**) The healing abutment is positioned.

**Figure 10 ijerph-15-02361-f010:**
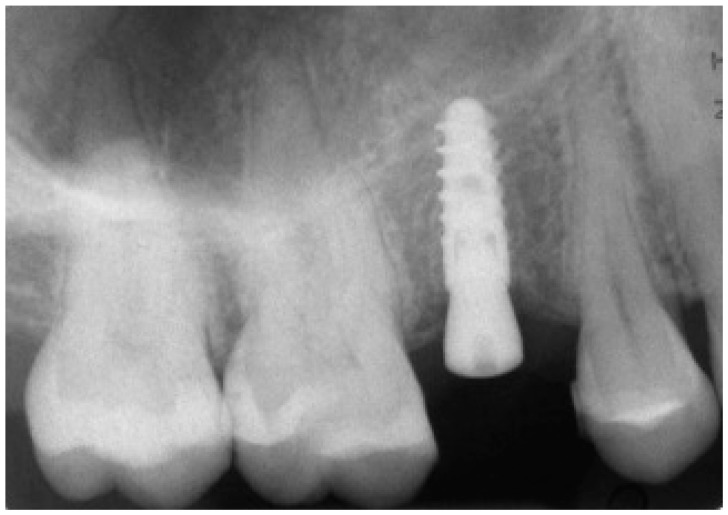
Post-surgical rx control of the implant and healing abutment.

**Figure 11 ijerph-15-02361-f011:**
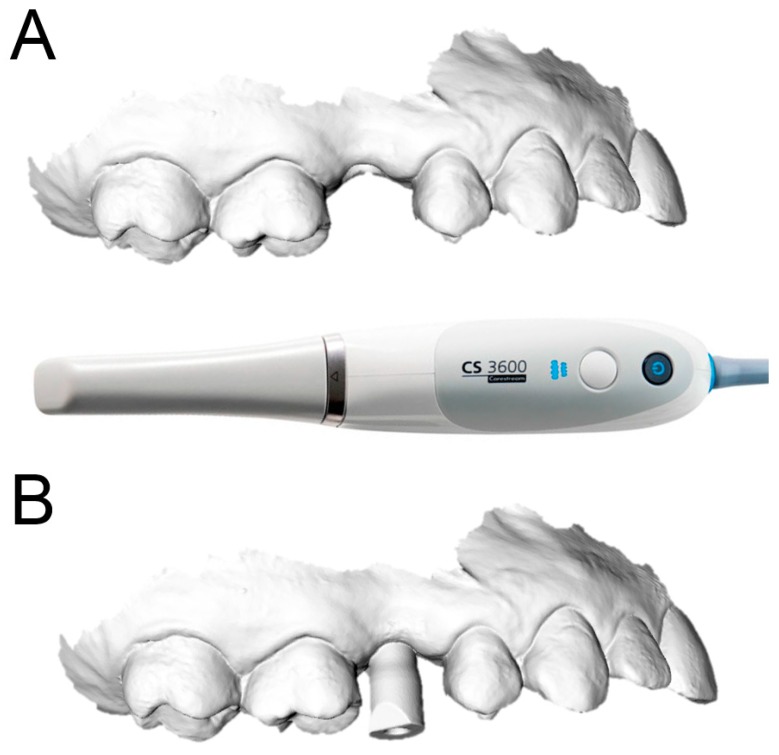
Optical impression with CS 3600^®^ (Carestream Dental, Atlanta, GA, USA). (**A**) Without scanbody. (**B**) With scanbody.

**Figure 12 ijerph-15-02361-f012:**
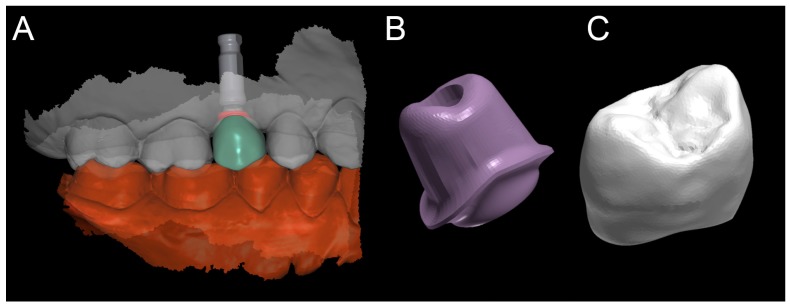
Computer assisted design project (EXOCAD^®^, Darmstadt, Germany). (**A**) CAD scene. (**B**) Details of the individual zirconia abutment (upper portion of the hybrid abutment); and (**C**) PMMA crown.

**Figure 13 ijerph-15-02361-f013:**
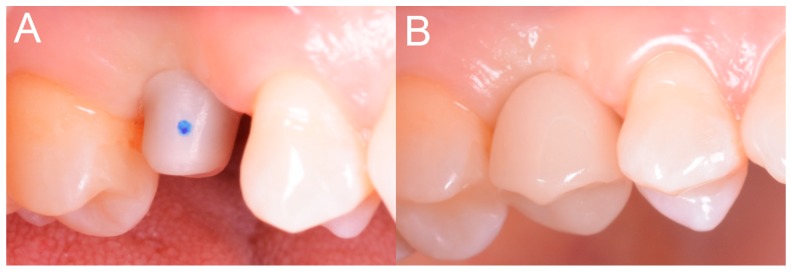
Delivery of the individual zirconia abutment (**A**) and the provisional PMMA crown (**B**).

**Figure 14 ijerph-15-02361-f014:**
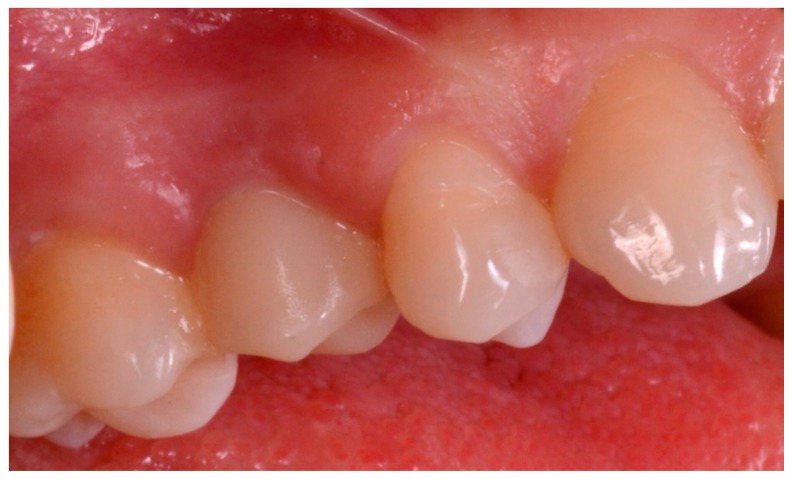
Delivery of the final monolithic translucent zirconia restoration.

**Figure 15 ijerph-15-02361-f015:**
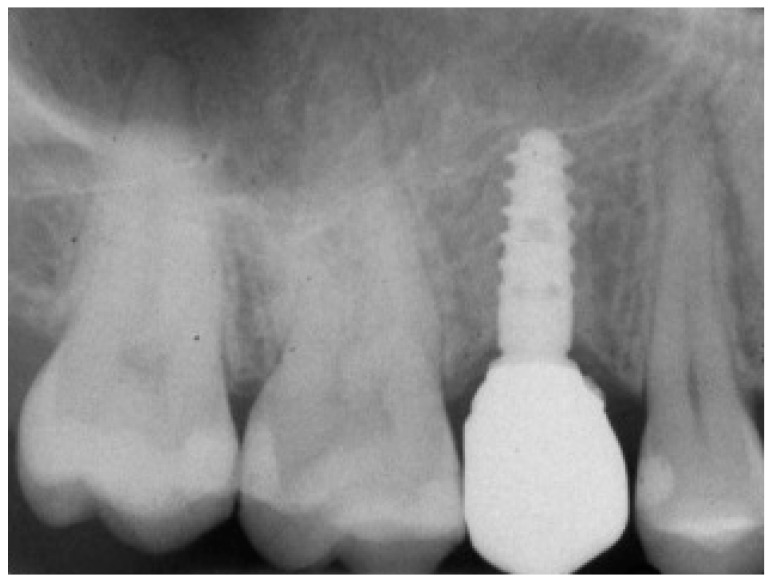
One-year follow-up rx control of the implant supported crown.

**Table 1 ijerph-15-02361-t001:** Distribution of the implants with related failures for stability and complications.

	N° of Planned Implants	Insufficient Stability at Placement	Immediate Post-Operative Complications	Complications During Follow-Up	Failures
*Site*					
Maxilla	19 (50.0%)	0/19 (0.0%)	0/19 (0.0%)	1/19	0/19
Mandible	19 (50.0%)	1/19 (5.2%)	0/19 (0.0%)	1/19	1/19
*Position*					
Incisors	3 (7.9%)	0/3 (0.0%)	0/3 (0.0%)	0/3	0/3
Cuspids	1 (2.6%)	0/1 (0.0%)	0/1 (0.0%)	0/1	0/1
Premolars	20 (52.6%)	0/20 (0.0%)	0/20 (0.0%)	1/20	0/20
Molars	14 (36.9%)	1/14 (7.1%)	0/13 (0.0%)	1/13	1/14
*Length*					
8.0 mm	14 (36.9%)	0/14 (0.0%)	0/0 (0.0%)	0/14	0/14
10.0 mm	18 (47.3%)	1/18 (5.5%)	0/0 (0.0%)	1/17	1/18
12.0 mm	6 (15.8%)	0/6 (0.0%)	0/0 (0.0%)	1/6	0/6
*Diameter*					
3.3 mm	7 (18.4%)	0/7 (0.0%)	0/0 (0.0%)	0/7	0/7
3.75 mm	14 (36.9%)	1/14 (7.1%)	0/0 (0.0%)	1/13	1/14
4.1 mm	9 (23.7%)	0/9 (0.0%)	0/0 (0.0%)	0/9	0/9
4.8 mm	8 (21.0%)	0/8 (0.0%)	0/0 (0.0%)	1/8	0/8
*Restoration*					
FPP	20 (55.6%)	0/20 (0.0%)	0/0 (0.0%)	0/20	0/20
SC	16 (44.4%)	1/16 (6.2%)	0/0 (0.0%)	2/15	1/16

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
