# Peer review of "Full in-Office Guided Surgery with Open Selective Tooth-Supported Templates: A Prospective Clinical Study on 20 Patients"

_ijerph, 2018, doi:10.3390/ijerph15112361_

Round 1

Reviewer 1 Report

very good research paper, excellent english form. very innovative topic that can help the readers and the clinicians to understand a new design

of templates for guided implant surgery (static guided surgery). my suggestion is to accept this paper after some minor modifications are

performed by the authors. only a minor discretional revision is required.

only a few modifications are required:

line 53, please remove "and not at all cut and dry", it is not needed here

line 82. there is a typing error, with a sentence in parenthesis, please remove parenthesis, it should read:

"For this reason, surgeons use these technologies only in the case of multiple implants and complex cases, in which the cost for the production of the guide is fully justified by the number of inserted fixtures."

line 121, it should read "CS 3600" (and not CS3600). correct please.

line 129, it should read "CS 9300" (not CS9300). please amend.

line 145, please erase the sentence of the costs, because you should not mention about costs here. please delete the following:

"The cost for the use of such software was limited: the basic platform of the prosthetic CAD software was purchased permanently at the price of €3500 + VAT, while the subscription to the software of surgical CAD (in cloud) cost €800 per year."

i ask you to remove this because costs are different in each countries and this sentence may be unappropriate, please cancel it. no detailed information about the costs.

line 165, there is a typing error, "positio" please correct into "position"

line 188, again please remove price details, and modify the last sentence into:

"with a variable price, in relation to the number of planned cases and exports requested).

line 308, "In the case, however, the fit was inadequate" please correct also punctuation

i do not see the table embedded in the pdf paper of the article, can you insert it in? 

line 472- 473 please remove the prices

line 475, there is a parenthesis that should be removed, please amend.

line 532, please eliminate the last sentence of the discussion and the related reference because it is not pertinent here.

finally, the authors should remember to embedd the figures into the pdf manuscript - if any - and all figures should have

proper citation throughout the paper.

Author Response

DEAR REVIEWER N°1 PLEASE FIND ATTACHED DETAILED REPLY TO ALL YOUR QUESTIONS AND CRITICISMS. YOU FIND ALL REPLIES TO YOUR QUESTIONS AND SUGGESTIONS IN CAPITAL LETTERS:

- very good research paper, excellent english form. very innovative topic that can help the readers and the clinicians to understand a new design of templates for guided implant surgery (static guided surgery). my suggestion is to accept this paper after some minor modifications are performed by the authors. only a minor discretional revision is required.

THANK YOU WE HAVE MODIFIED OUR PAPER ACCORDING TO YOUR SUGGESTIONS AND WE HAVE UPLOADED THE REVISED VERSION ACCORDINGLY. ALL MODIFICATIONS HAVE BEEN MADE DIRECTLY IN THE TEXT AND HAVE BEEN HIGHLIGHTED USING A DIFFERENT COLOUR (YELLOW). 

only a few modifications are required:

WE HAVE PERFORMED THEM ALL

line 53, please remove "and not at all cut and dry", it is not needed here

WE HAVE REMOVE IT AS PER YOUR SUGGESTION

line 82. there is a typing error, with a sentence in parenthesis, please remove parenthesis, it should read:

"For this reason, surgeons use these technologies only in the case of multiple implants and complex cases, in which the cost for the production of the guide is fully justified by the number of inserted fixtures."

WE HAVE CORRECTED IT. 

line 121, it should read "CS 3600" (and not CS3600). correct please.

WE HAVE CORRECTED IT.

line 129, it should read "CS 9300" (not CS9300). please amend.

WE HAVE CORRECTED IT.

line 145, please erase the sentence of the costs, because you should not mention about costs here. please delete the following:

"The cost for the use of such software was limited: the basic platform of the prosthetic CAD software was purchased permanently at the price of €3500 + VAT, while the subscription to the software of surgical CAD (in cloud) cost €800 per year."

i ask you to remove this because costs are different in each countries and this sentence may be unappropriate, please cancel it. no detailed information about the costs.

WE HAVE REMOVED THIS SENTENCE ACCORDINGLY

line 165, there is a typing error, "positio" please correct into "position"

WE HAVE CORRECTED IT.

line 188, again please remove price details, and modify the last sentence into:

"with a variable price, in relation to the number of planned cases and exports requested"

WE HAVE ERASED ALL REFERENCES TO COSTS.

line 308, "In the case, however, the fit was inadequate" please correct also punctuation

WE HAVE CORRECTED PUNCTUATION

i do not see the table embedded in the pdf paper of the article, can you insert it in? 

WE HAVE EMBEDDED THE TABLE IN THE ARTICLE AS REQUESTED. ORIGINALLY WE HAD UPLOADED IT AS A SEPARATE FILE, NOW WE ADDED IT IN THE TEXT.

line 472- 473 please remove the prices

WE REMOVED THE PRICES ALSO FROM HERE

line 475, there is a parenthesis that should be removed, please amend.

WE HAVE REMOVED IT.

line 532, please eliminate the last sentence of the discussion and the related reference because it is not pertinent here.

WE HAVE ERASED THE LAST SENTENCE AND THE RELATED REFERENCE.

finally, the authors should remember to embedd the figures into the pdf manuscript - if any - and all figures should have proper citation throughout the paper.

WE HAVE EMBEDDED ALL FIGURES INSIDE THE MANUSCRIPT, WITH PROPER LEGENDS.. ORIGINALLY WE HAD UPLOADED ALL FIGURES AS SEPARATE FILES BUT MAYBE THERE WAS SOME PROBLEMS IN THE UPLOADING INSIDE THE TEXT WE APOLOGIZE FOR THAT.

Reviewer 2 Report

excellent work well written and with high quality structure, that deals innovation for a new chairside methodology.

my only suggestion would be to remove any reference to the price of the materials and methodologies, however

i suggest the publication of the present study in its current form.

Author Response

excellent work well written and with high quality structure, that deals innovation for a new chairside methodology.

my only suggestion would be to remove any reference to the price of the materials and methodologies, however

i suggest the publication of the present study in its current form.

THANK YOU VERY MUCH, WE HAVE MODIFIED THE TEXT IN ACCORDANCE WITH YOUR SUGGESTIONS, REGARDS.